# Acid Mine Drainage Effects in the Hydrobiology of Freshwater Streams from Three Mining Areas (SW Portugal): A Statistical Approach

**DOI:** 10.3390/ijerph191710810

**Published:** 2022-08-30

**Authors:** Ana Teresa Luís, José Antonio Grande, Nuno Durães, María Santisteban, Ángel Mariano Rodríguez-Pérez, Eduardo Ferreira da Silva

**Affiliations:** 1Department of Geosciences, GeoBioTec—Geobiosciences, Geotechnologies and Geoengineering Research Center, Campus de Santiago, University of Aveiro, 3810-193 Aveiro, Portugal; 2Department of Mining, Mechanic, Energetic and Construction Engineering, Higher Technical School of Engineering, University of Huelva, 21007 Huelva, Spain

**Keywords:** acid mine drainage (AMD), Aljustrel, Lousal, S. Domingos, Iberian Pyrite Belt (IPB), diatoms, surface waters, mines

## Abstract

Aljustrel, Lousal and S. Domingos mines are located in the Iberian Pyrite Belt (IPB), one of the greatest massive sulfide ore deposits worldwide. These mines’ surrounding streams are affected by Acid Mine Drainage (AMD). The main purpose of this study was to understand AMD influence in the water quality and diatom behavior. Thus, waters and diatoms were sampled in 6 sites from the 3 selected mines on winter and summer of 2016. The highest concentrations were found in acidic sites: A3 (Aljustrel—Al, Cd, Cu, Fe and Zn (and lowest pH)) and L1 (Lousal—As, Mn, Ca, Mg, SO_4_^2−^ and conductivity). The most abundant diatom species was *Pinnularia aljustrelica* with 100% of dominance in A3 and S1 acidic sites, which puts in evidence this species adaptation to AMD harsh conditions. Multivariate cluster analysis allowed us to reinforce results from previous studies, where spatial differences were more relevant than seasonal ones. In 12 years (2004–2016), and with many transformations undertaken (re-opening and rehabilitation), there is a conservative behavior in the biological species (diatoms) and physicochemical concentrations (metals, pH and sulfates) from these three mining sites. This type of biogeochemical diagnosis is necessary for the sustainable use of these waters and the prevention of the polluting process, aimed to protect the water ecosystem and its biodiversity.

## 1. Introduction

The mining impact on the environment varies greatly depending on the methods employed to exploit the mineral deposits, the type and extension of the mineral resources, the waste management and adequate pollution control and proper reclamation.

The Iberian Pyrite Belt (IPB) is in the SW sector of the Iberian Massive, comprising part of Portugal (Alentejo) and of the provinces of Huelva and Seville (SW Spain). It forms an area of about 240 km long and 35 km wide. Aljustrel, Lousal and S. Domingos are three Portuguese mines from the Iberian Pyrite Belt (IPB) presenting massive sulfide deposits. These mines are now closed (some Aljustrel deposit masses are being explored), but the potential toxic elements enriched tailings remain there. Some remediation works are being performed nowadays to mitigate pollution, such as re-vegetation and tailings cover, but none of them seem to be very effective [1].

The Acid Mine Drainage (AMD) phenomenon is the main factor responsible for the modification of the physicochemical characteristics of the natural waters, when sulfide minerals from tailings are exposed to atmospheric, hydrological and biological weathering (oxygen, water and chemoautotrophic bacteria) becoming oxidized and resulting in low pH, sometimes achieving negative values [2,3]. The low alkalinity, very high electrical conductivity [4,5] and high contents of dissolved metal(loid)s and SO_4_^2−^ [6,7,8] are also typical of AMD affected waters that sometimes drain to dams for public supply [9] and, also, to the ocean [10].

AMD effects on aquatic ecosystems are twofold: (a) impacted communities experience lethal levels of pH and metal(loid)s, which lead to a decrease in algal species richness and diversity [11,12]; (b) communities are restricted to tolerant organisms, which are able to survive in these conditions. Alterations in nutrient cycles and abiotic changes are experienced by these communities, with large impact in biotic relationships including extinction and succession of species and groups of sensitive taxa [13]. Diatom’s metal response models are difficult to establish because metal(loid) contamination is frequently associated with acidic environments. Several studies in metal-polluted rivers have shown that diatoms respond to perturbation, not only at the community level through shifts in dominant taxa [14,15,16,17], but also with changes in diversity [15,16,17,18].

Therefore, the aim of this work is to characterize the water chemistry and diatom communities (hydrobiology) from AMD impacted streams of Aljustrel, Lousal and S. Domingos mining areas. As a secondary objective, to identify the mineralogical sources affecting water chemistry and diatoms.

## 2. Study Area

### 2.1. Geology

Aljustrel, Lousal and S. Domingos mining areas are located in the South of Portugal and belong to the Iberian Pyrite Belt (IPB) (Figure 1), one of the most important and larger metallogenic provinces of the world, with an area of about 230 km long and 30 to 60 km wide, extending from Portugal to Spain. The IPB is characterized by the occurrence of volcanic-associated massive sulfide deposits (VMS-type) and some Mn mineralizations [19,20]. Three main and common geological units are part of the IPB: a substrate composed by phyllite and quartzite rocks, overlaid by the Volcano-Sedimentary Complex (VSC) that hosts the mineralization and the Culm (flysch) group [19].

### 2.2. Mineralisation

The mineralization of Aljustrel is distributed by six important mineral masses (Moinho, Feitais, Estação, Gavião, Algares and São João) hosted in the VMS unit, and composed predominantly by pyrite (FeS_2_) and minor galena (PbS), sphalerite (ZnS), arsenopyrite (FeAsS), antimony and sulfosalts [21].

The Lousal ore deposit hosted 18 mineralized sulfides lenses, which also include the one from the old Caveira pyrite mine, mainly composed by pyrite and minor chalcopyrite (CuFeS_2_), galena, sphalerite, pyrrhotite (Fe_(1−X)_S), marcasite (FeS_2_), bournonite (CuPbSbS_2_), tetrahedrite (Cu_12_Sb4S_13_), arsenopyrite, cobaltite (CoAsS), magnetite (Fe_2_O_4_) and native gold (Au) [22]. The paleogeographic environment of the Lousal deposit is similar to other massive sulfide deposits of the IPB related with black shales such as Montinho (Portugal), Tharsis and Sotiel (Spain) [20]. Lousal presents carbonated lenses, carbonated ore matrix and cobalt bearing minerals in the deposits [23].

S. Domingos mine is located in the southeast of Portugal, at Mértola municipality (Figure 1). This ore deposit consists of a single subvertical massive pyrite body [24] of 27 Mt, with variables quantities of chalcopyrite, galena and sphalerite [25] with a huge surficial gossan overlying the mineralization.

### 2.3. Exploitation

These mines have a long exploitation history, dating back to pre-Roman periods [24]. After centuries of almost complete inactivity, the mines were again worked during the XIX and XX centuries, focusing mainly on the production of copper and sulfuric acid. Mining activity has intensively worked the base metals present in massive ores and produced copper, zinc, zinc + lead and tin ore concentrates until the present time. It was in the XX century, until the mid-80s–90s that these mines experienced their greatest development, except for S. Domingos whose exploitation ceased first, in 1966 due to the reserves exhaustion. Since 2009, mining was re-opened in Aljustrel for the exploitation of Zn and Cu concentrates.

The sulfides’ exploitation led to the production of high amounts of waste rock dams that still remain nowadays in the three mining sites, occupying large areas in the mine’s surrounding. Slags (particularly the modern ones), smelting ashes and pyrite-rich samples are the mine wastes remaining that represent the main sources of Acid Mine Drainage (AMD) [27].

## 3. Methods

### 3.1. Sample Collection and Processing

Sampling of surface water, salt efflorescences and diatom communities were carried out in 6 sites selected in the 3 mines (L1 in Lousal; A1, A2 and A3 in Aljustrel; S1 and S2 in S. Domingos) on winter and summer of 2016. To determine physical and chemical parameters of surface waters, a volume of 1 L was collected in acid-rinsed polyethylene bottles. Temperature (T °C), pH, electrical conductivity (Cond; at 25 °C) and oxidation-reduction potential (ORP) were recorded on site, using a multiprobe WTW Multiline P4 SET. Water samples were then returned in a cool box to the laboratory.

In the lab, a volume of 250 mL was taken from each sample and filtered through 0.45 μm Millipore membrane filters using an all-plastic pressurized filtration system. A sub-sample of these filtered waters, was preserved with ultra-pure nitric acid (HNO3) to prevent metal precipitation and bacterial growth, was used for trace and major cation analysis, and another portion, unacidified, for anion analysis. Samples were then stored at 4 °C until being analyzed.

The epipsammic diatom samples were collected by removing the top layer of the sediment surface with a syringe. Two samples were taken, one kept alive and the other was preserved with formalin solution (5% final concentration). Following the sampling protocol [28], pools of stagnant water and shaded sites were avoided.

### 3.2. Analytical Methods of Waters, Salt Efflorescences and Diatoms

Sulfate concentration was determined by ion chromatography (Dionex 1000i ion chromatograph with a Dionex AS4-SC column) and major elements and metal(loid)s (Al, As, Ca, Cd, Cl, Cu, Fe, Mg, Mn, Na and Zn) were analyzed by ICP-MS in the Laboratory of Geochemistry at the Geosciences Department of the University of Aveiro. A rigorous quality control program was implemented during water chemical analysis, which included reagent blanks and duplicate samples. The precision and bias error of the chemical analysis were less than 10%.

X-ray Powder Diffraction (XRD) analysis of salt efflorescences were carried out in the Geosciences Department of the University of Aveiro using a X’Pert MPD (Almelo, The Netherlands) equipped with an automatic divergence slit, CuKα (λ = 1.5405 Å) radiation (20 mA and 40 kV) and a Ni filter, in the range of 4–70° (2θ°), through a counting step of 0.02° per s.

Live diatom samples were examined to exclude the possibility of the presence of dead diatoms in order to avoid abundance errors. From the other set of samples (preserved with formalin), an aliquot (after cleaning off formalin) was treated with HNO3 (65%) and potassium dichromate (K_2_Cr_2_O_7_) at room temperature for 24 h, followed by three centrifugations (1500 r/min) to wash off the excess of acid. Then, permanent slides with clean diatoms were prepared using Naphrax^®^ resine.

Diatoms were identified and semi-quantified (400 valves per sample) under a light microscope (Leitz Biomed 20 EB) using a 100× objective (N.A. 1.32). Taxonomy was based on the floras of [28,29,30,31,32]. 

### 3.3. Data Analysis

The final data matrix was composed by 17 variables: Nº Sps (species number), %PALJ (percentage of *Pinnularia aljustrelica*), Al, As, Ca, Cd, Cl, Cu, Fe, Mg, Mn, Na, Zn, SO_4_^2−^, pH, ORP and Cond, being the first two biotic and the other 15 abiotic variables. Data processing was performed based on this matrix to obtain samples and variables clusters, using the software package Statgraphics Centurion XVI (Statgraphics Technologies, Inc., Virginia, USA). The Ward method or ‘second-order central moment’ was applied, which is a hierarchical method that calculates the mean of all the variables for each cluster; after, it calculates the Euclidean distance between each factor and the average of its group and then adds the distances from each case. In each step, the clusters that are formed are those that yield the smallest increment in the total sum of the intracluster distances [4,33]. The application of this technique, in this study, helped to classify the variables into different ‘categories’ as follows. The linear cluster of the hydro-chemical–biological variables and the linear cluster of the sampling sites, were obtained with non-standardized/adjusted data (just homogenized to mg/L). When applied to a set of variables, cluster analysis orders and classifies them in the most homogeneous groups possible based on the similarity of the variables themselves [4], and then, to propose a hydrochemical operation model, based on the grouping of variables in different subclusters, which are related with the physicochemical characteristics of the waters.

## 4. Results and Discussion

The results of the biotic (2 variables: Number of Species (Nº Sp) and % of *Pinnularia aljustrelica* (%PALJ)) and physicochemical (15 variables) parameters (total of 17 variables) from the six sampling sites (summer and winter campaigns of 2016) are present in Table 1.

Two main types of waters can be distinguished just based on pH values: sites A1 and A2 in Aljustrel showing alkaline conditions were separated from the remaining sites with acid pH. Highest pH and highest Cl and Na concentrations were obtained in A1, due to halite dissolution [17]. In addition, highest diatom diversity was found here (36 Nº Sp), indicating more suitable conditions of these microalgae to live. Regarding the acidic sites, the lowest pH was registered in S1 (pH 2.19), but very close to the value found in site A3 (pH 2.23). The highest Al, Cd, Cu, Fe and Zn concentrations were registered also in A3. It is here that the harsh conditions of high metals and acidity are more evident [17]. Moreover, in the past, samples from site A3 had the highest Al, Cd, Cu and Fe concentrations (Table 2) [16,26].

Conductivity and sulfates in AMD contaminated waters go together because conductivity is directly and strongly influenced by the high sulfate ion concentration in this type of contaminated waters, especially in L1 (2016 campaigns; Table 1). Arsenic, Ca, Mg and Mn present the highest concentrations also in L1. With the exception of As, also in the past (2004 campaign; [34]), Ca, Mg and Mn showed the highest concentrations in Lousal samples (L1) when compared with Aljustrel (A3) and S. Domingos (S1) sites (Table 2). This points out to a greater contribution of host rock minerals dissolution (e.g., plagioclase, ferromagnesian silicates or even few carbonates) in the waters of L1, which can be due to a greater abundance and/or dissolution degree (buffer) of these minerals in Lousal.

Although all the analyzed samples were clearly affected by AMD (excepting A1 and A2), exhibiting pH values below three, high conductivity and high sulfate and iron concentrations, there were differences in the geochemical signatures that allow their individualization. Therefore, cluster analysis was conducted to further validate the first observations made on Table 1 and Table 2. With a same data matrix (input), from 2016, two clusters (output) were obtained: cluster of samples (Figure 2) and cluster of variables (Figure 3).

The Lousal spatial differentiation is very evident in the cluster of samples (Figure 2), where it is located far away from the rest of the sites in a separated sub-cluster, on the far-right end, which is probably due to its high Mg, Ca and Mn concentrations that allow the individualization of a sub-cluster in the dendrogram of elements (Figure 3). In particular, Mg showed very high concentrations in L1 comparing to A3 and S1, explaining the prevalence of Mg-sulfates, as epsomite and hexahydrite precipitated here (Figure 4), while in the other sites from Aljustrel and S. Domingos melanterite dominated in the efflorescences [35]. These Mg efflorescences were found in other IPB’s Spanish mines (Herrerías, Riotinto, Lagunazo, Tharsis and S. Telmo) [36].

Acidic sites from Aljustrel (A3) and S. Domingos (S1, S2) are placed together near Lousal acidic site, but in a separated sub-cluster, on the left of the same cluster.

The first sub-cluster of Figure 3, at the far-left end, grouping together Nº Sp, pH, Cl and Na reveals the close relationship between these variables. The Na and Cl association indicates salts dissolution (halite) occurring in the Cenozoic sediments of Sado Basin [17,19] in Aljustrel, in sites A1 and A2 with higher concentrations in the most alkaline sites. The pH here is closely related with the diversity (Nº Sp) of diatom communities (>pH > Nº Sp). This fact can indicate a dependent relationship between the high diatom species number and the high chlorinity of the medium, classifying these waters as *Circumneutral-Na-Cl* type: waters not affected by mining activity with a circumneutral to alkaline pH and high electrical conductivity values [17].

The *Acid–metal–sulfate* group on the second cluster (composed by two sub-clusters: one on the middle and another on the far right end of the Figure 3) comprise the remaining parameters. This cluster reflects the AMD contamination, characterized by high metals and SO_4_^2−^ contents, and materialized by very high conductivities, since sulfuric acid generation through mining tailings sulfide’s oxidation provoke an abrupt decrease in pH to values close to 2 (2.19–2.59), as well as ore and hosting rocks minerals progressive dissolution (high amounts of trace metal(loid)s from ore and Al, Ca and Mg from the hosting rocks).

An important aspect of this cluster (see sub-cluster on the middle of the figure) is the association of metals (especially Al, Cd, Cu and Zn) with *P. aljustrelica* (Figure 3), a well-adapted species to this type of contaminated waters. 

Conductivity and SO_4_^2−^ are grouped together (sub-cluster on the right of the second cluster), as already referred above, especially for L1 because in acid–metal contaminated waters this anion is the main contributor to the high ionic content [5,37]. Fe and ORP go together also in this group. Fe is in present in very high concentrations and undergoes hydrolysis which is crucial for the formation of AMD. These are oxidizing systems (positive ORP), thus as higher is the oxidizing capacity of the waters, higher the ORP is.

On the other hand, the main sources of Mn and Mg are rocks from the VSC, particularly the jaspers, the tuffs and the volcanic rocks that can be present in almost every mines of the IPB, but here are more evident in Lousal (L1).

## 5. Conclusions

Multivariate analysis, i.e., cluster analysis, has been confirmed as a useful tool for characterizing AMD processes by grouping physicochemical variables, according to degree of importance, as well as the samples/sites by grade of physicochemical similarity.

Site L1 (Lousal mining area) has a very high conservative behavior regarding Ca, Mg and Mn as shown in previous works, and, also, in this one. Secondary Mg-sulfate soluble salts (as epsomite and hexahydrite) were found in Lousal (site L1) but not in the other Portuguese (Aljustrel and S. Domingos) IPB mines, reflecting the higher Mg concentration of AMD in Lousal. 

Site A3 (Aljustrel mining area) is the most affected by harsh AMD conditions of low pH, high sulfates, conductivity and metals concentrations (Al, Cd, Cu, Fe and Zn).

The new results of this study (from 2016) allowed to confirm and validate previous considerations [15,16,17,34,38,39] regarding spatial variation of biological and physico-chemical parameters that is more important than seasonal one. Parameters show more considerable changes from site to site than from season to season in a same site.

In a period of 12 years (2004–2016) and with many transformations undertaken, such as mine re-opening (Aljustrel) and rehabilitation works (Aljustrel, Lousal and S. Domingos), there is a conservative behavior in the biological (dominating *Pinnularia aljustrelica* acidic species in AMD affected sites) and physicochemical parameters (very high sulfates, conductivity and metals concentrations and low pH) of the three mining sites.

Further regarding this topic of biogeochemistry of AMD affected waters in all IPB (Spanish and Portuguese parts), this working team has been developing many studies in the last years: [40,41,42,43,44], reinforcing the importance of diatoms as bioindicators, that can assess environmental changes over time, evaluating the contamination impact of these AMD peculiar systems, helping in the decision of the most appropriate intervention, with economic and environmental costs reduction, contributing immediately to the quality improvement of waters. Thus, this type of biogeochemical diagnosis is necessary for the sustainable use of these waters and the prevention of the polluting process, aimed to protect the water ecosystem and its biodiversity from a circular economy perspective that can make possible the reconciliation of mining activity with the use of these waters for non-mining purposes. 

## Figures and Tables

**Figure 1 ijerph-19-10810-f001:**
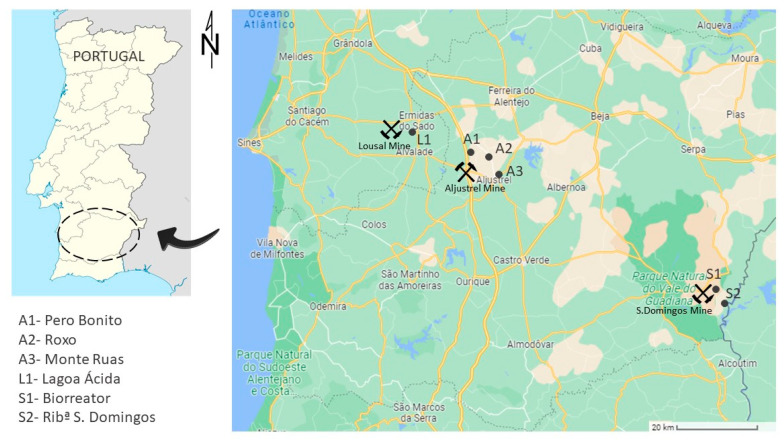
Location of Lousal, Aljustrel and S. Domingos mines, and, respective, sampling points L1—Lagoa Ácida, A1—Pero Bonito, A2—Roxo, A3—Monte Ruas, S1—Biorreator and S2—Ribª S. Domingos; names of sites are given to facilitate comparison with the same sites of author’s previous papers [15,16,17,26].

**Figure 2 ijerph-19-10810-f002:**
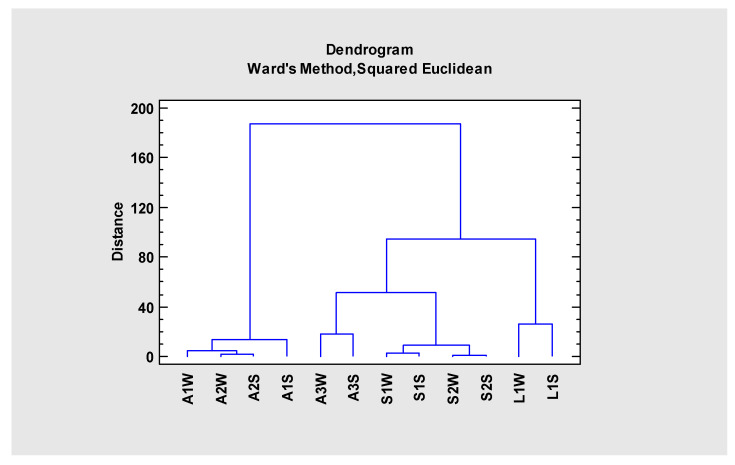
Dendrogram resulting from the cluster of the sampling sites using the Ward, Euclidean Square Method.

**Figure 3 ijerph-19-10810-f003:**
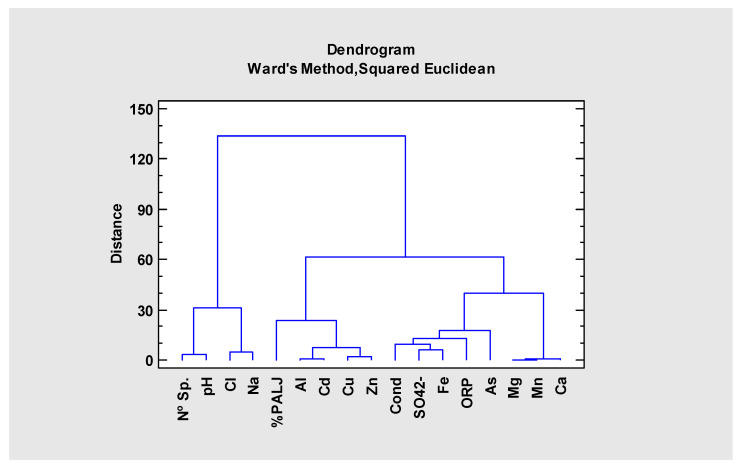
Dendrogram resulting from the cluster of the physicochemical and biological variables using the Ward, Euclidean Square Method.

**Figure 4 ijerph-19-10810-f004:**
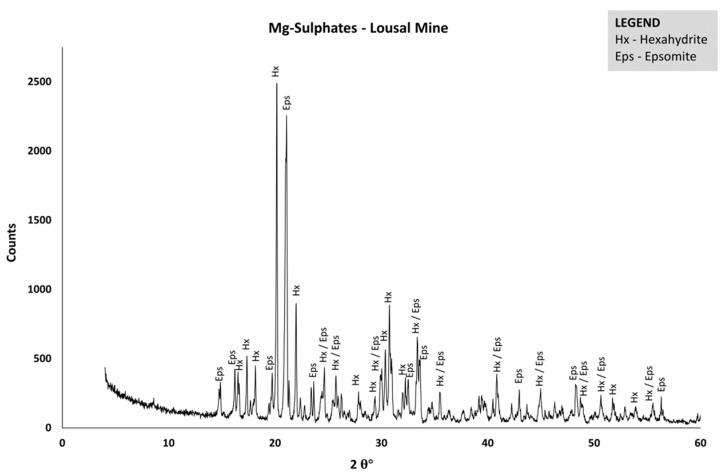
X-ray diffractogram showing the dominance of Mg-sulfates (epsomite and hexahydrite) on site L1.

**Table 1 ijerph-19-10810-t001:** Biological and physicochemical parameters (concentrations in mg/L) from the selected points in 2016. Conductivity measured in µS/cm and ORP in mV; color code for highest concentrations of the parameters in the sampling sites: A1, A2 blue; S1, S2 green; A3 orange; L1—purple; the letter at the end of sample name indicates the sampling period, thus W—winter and S—summer; (N.D.—not detectable).

Sites	Nº Sp	%PALJ	pH	Cond	ORP	SO_4_^2−^	Cl	Na	Mg	Ca	Al	As	Cd	Cu	Fe	Mn	Zn
A1W	19	N.D.	8.70	2184	99	117	516	257	106	153	0.01	N.D.	N.D.	N.D.	0.15	0.05	N.D.
A2W	19	N.D.	8.22	1518	120	55	313	172	65	104	0.01	N.D.	N.D.	0.01	0.10	0.05	N.D.
A3W	9	41.39	2.59	5200	498	3901	170	94	221	208	275.57	0.54	0.29	53.77	4583.94	30.96	128.62
S1W	1	100	2.56	3451	522	1867	98	63	95	96	146.18	0.36	0.15	16.04	956.49	10.60	25.14
S2W	4	30.67	2.55	5064	498	1469	96	71	79	81	94.09	0.15	0.8	10.49	921.93	6.74	15.52
L1W	3	87.28	2.51	8300	502	5192	288	260	902	585	167.20	1.36	0.11	11.46	3897.71	117.63	60.69
A1S	34	N.D.	8.11	3153	303	217	698	275	130	238	0.01	0.01	N.D.	N.D.	0.15	N.D.	0.03
A2S	36	N.D.	8.02	1488	161	51	370	152	58	111	0.02	N.D.	N.D.	N.D.	0.51	0.029	0.03
A3S	1	100	2.23	6426	460	3739	403	158	233	237	199.20	0.09	0.25	58.24	2038.37	27.38	117.36
S1S	1	100	2.41	3656	300	2229	109	58	96	112	161.94	0.35	0.16	20.70	1153.48	11.45	34.67
S2S	4	30.75	2.19	4413	381	2006	121	66	93	118	130.53	0.02	0.10	16.17	1189.42	9.65	27.04
L1S	2	23.41	2.54	7782	475	2007	390	218	726	514	115.81	0.23	0.10	12.27	3463.86	104.95	81.02

**Table 2 ijerph-19-10810-t002:** Physicochemical parameters and their concentrations (mg/L) in the acid sampling points L1, A3 and S1 in previous studies (summer and winter of 2004 [34], 2008 [16] and 2014 (unpublished results), respectively) and in this study (summer and winter of 2016) for comparison. Conductivity was measured in µS/cm. (color code for highest concentrations of the parameters in the acidic sampling sites: A3 orange; L1—purple; S1 green).

Sites	L1				A3				S1			
	Summer (2004)	Winter (2004)	Summer (2016)	Winter (2016)	Summer (2008)	Winter (2008)	Summer (2016)	Winter (2016)	Summer (2014)	Winter (2014)	Summer (2016)	Winter (2016)
pH	2.4	2.8	2.5	2.5	2.3	2.0	2.2	2.6	2.6	2.4	2.4	2.6
Cond	8970	6650	8300	7782	14,200	14,100	6426	5200	3806	3690	3656	3451
SO_4_^2−^	8109	6069	2007	5192	16,995	36,900	3739	3901	2323	1960	2229	1867
Cl	539	432	390	288	99	1	403	170	234	108	109	98
Ca	568	453	514	586	250	434	237	208	61	81	112	96
Mg	1061	743	726	902	438	365	233	221	58	69	96	95
Na	297	227	218	261	88	40	158	94	37	35	58	63
Al	63.0	35.0	115.8	167.2	757.0	482.0	199.2	275.6	211.0	295.0	161.9	146.2
As	0.0	0.0	0.2	1.4	29.8	52.0	0.1	0.5	0.3	0.6	0.3	0.4
Cd	0.3	0.2	0.1	0.1	1.7	2.1	0.3	0.3	0.1	0.1	0.2	0.2
Cu	3.7	2.5	11.5	11.5	130.0	216.0	58.2	53.8	21.0	10.7	20.7	16.0
Fe	562.0	504.0	3463.9	3897.7	3688.0	4841.0	2038.4	4583.9	178.0	81.0	1153.5	956.5
Mn	157.0	110.0	105.0	117.6	89.0	66.0	27.4	31.0	11.3	5.6	11.5	10.6
Zn	115.0	77.0	81.0	60.7	970.0	1112.0	117.4	128.6	38.0	16.8	34.7	25.1

## Data Availability

Not applicable.

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
