# Peer review of "Acid Mine Drainage Effects in the Hydrobiology of Freshwater Streams from Three Mining Areas (SW Portugal): A Statistical Approach"

_ijerph, 2022, doi:10.3390/ijerph191710810_

Round 1

Reviewer 1 Report

IJERPH-1863686

Acid Mine Drainage effects in the hydrobiology of freshwater streams from three mining areas (SW Portugal): a statistical approach, BY Ana Teresa Luís, José Antonio Grande, Nuno Durães, María Santisteban, Ángel Mariano Rodríguez-Pérez and Eduardo Ferreira da Silva

This work describes several sites affected by AMD along the Iberian Pyrite Belt in Portugal, mainly physico-chemical parameters and diversity of diatoms. Low pH along with high electric conductivity, sulfate and trace element concentrations were consistent with high impact from AMD. The use of cluster analysis to understand AMD was claimed as a main contribution of this work.

Sites A1 and A2 are obviously very different from the other sites. I could not find in the text if they are, or were in the past, affected by AMD in some way. Another issue is how are these sites related to A3 (A3 shows obvious effects of AMD). Can sites A1 and A2 be considered “unaffected by AMD”?

A third issue is the general consideration of the behavior of several parameters as “conservative” despite differences in values. Are these differences significant, or not?

Minor issues:

Lines 17-18: please re-write this sentence.

Line 24: species adaptation to what?

Line 29: what do you mean as “general and conservative”?

Lines: 47-53: Please split this sentence.

Line 55: do AMD impacts only the algal communities? What about animals and heterotrophic microorganisms?

Lines 57-59: I could not understand this sentence.

Line 77: what do you mean as VMS? Is it the VSC mentioned in line 264?

Lines 188-192: Please

Line 192: suitable conditions?

Table 1: Please use N.D. (not detected) or the like instead of 0, since “0” here means “below the detectable/quantifiable level”. Please put subscript in sulfate. I think it would be easier to analyze the table if samples were grouped by place, instead of being grouped by season. It would be useful to mention in the caption that data were obtained from samples collected in 2016.

Table 2: Is it possible to provide 1-2 sentences in the Methods section to explain how the 2014 data were obtained, and put them as results from the present work instead of “unpublished”? Please delete the “b.d.l.- below detection limit”, since it was not used in the table. It probably wold be more useful in Table 1.

Lines 217-218: Sites A1 and A2 does not seem to be affected by AMD.

Line 223: According to figure 2, the differences between A1 and A2 on one side, and the other samples at the other were even higher, but not mentioned in the text.

Figures 2-3: Where they build with data obtained from the 2016 samples?

Lines 258-259: Please re-write the explanation of what cluster is discussed.

Lines 261-263: in figure 3, Fe is grouped together with sulfate and conductivity. Please include Fe in the interpretation of clustering.

Line 264: Please explain the meaning of VSC the first time it appears.

Line 275-276: Table 2 shows obviously different values along time for some parameters, for instance Al, Cu and Fe (L1); conductivity, sulfate, Al, As, Cd, Cu, Mn and Zn for A3; and Fe for S1. Why these differences were not discussed? Can they be explained by changes in the operation of the mining sites?

Lines 282-284: Please re-write this sentence.

Lines 286-289: this paragraph repeats the comment on lines 275-276. The comment for lines 275-276 can be applied to this paragraph as well.

Line 288: biological data was provided only for 2016. Thus, there is no data in the present work to conclude anything about biological parameters along time.

Author Response

In the manuscript file, all the alterations are underlined in yellow

IJERPH-1863686

Acid Mine Drainage effects in the hydrobiology of freshwater streams from three mining areas (SW Portugal): a statistical approach, BY Ana Teresa Luís, José Antonio Grande, Nuno Durães, María Santisteban, Ángel Mariano Rodríguez-Pérez and Eduardo Ferreira da Silva

Reviewer 1

This work describes several sites affected by AMD along the Iberian Pyrite Belt in Portugal, mainly physico-chemical parameters and diversity of diatoms. Low pH along with high electric conductivity, sulfate and trace element concentrations were consistent with high impact from AMD. The use of cluster analysis to understand AMD was claimed as a main contribution of this work.

Sites A1 and A2 are obviously very different from the other sites. I could not find in the text if they are, or were in the past, affected by AMD in some way. Another issue is how are these sites related to A3 (A3 shows obvious effects of AMD). Can sites A1 and A2 be considered “unaffected by AMD”? Answer: Yes, sites A1 and A2 are unaffected by AMD. From our knowledge, never affected by AMD. They were considered in this study because they represent Na and Cl association indicating salts dissolution (halite) occurring in the Cenozoic sediments of Sado Basin.

A third issue is the general consideration of the behavior of several parameters as “conservative” despite differences in values. Are these differences significant, or not? Answer: Conservative along time in a same site, and also conservative between AMD contaminated sites despite some diferences in concentrations.

Minor issues:

Lines 17-18: please re-write this sentence. Answer: Done

Line 24: species adaptation to what? Answer: … to AMD harsh conditions. Lines 24-25

Line 29: what do you mean as “general and conservative”? Answer: sentence was re-written. Please see it now: “… there is a conservative behaviour in the biological species (diatoms) and physicochemical parameters of these three mining sites.” Lines 27-29.

Lines: 47-53: Please split this sentence. Answer: Done.

Line 55: do AMD impacts only the algal communities? What about animals and heterotrophic microorganisms? Answer: Animals are restricted to some micro-macroinvertebrates (for more information about life in AMD polluted environments please read Luís et al 2022 (reference nº 44)

Lines 57-59: I could not understand this sentence. Answer:

This sentence means that: Alterations in nutrient cycles and abiotic changes have a large impact in biotic relations as for example extinction of groups of sensitive taxa and predominance of adapted species to AMD environments. Sentence were re-written. Lines 60-62.

Line 77: what do you mean as VMS? Answer: line 78: volcanic-associated massive sulphide deposits.They occur within volcanosedimentary stratigraphic successions, and are commonly coeval and coincident with volcanic rocks. So, VSC is the (Vulcano Sedimentary Complex) where VMS occur. Is it the VSC mentioned in line 264? Answer: Yes.

Lines 188-192: Please Answer: ?? Missing sentence from the reviewer.

Line 192: suitable conditions? Done: changed for suitable. Line 195.

Table 1: Please use N.D. (not detected) or the like instead of 0, since “0” here means “below the detectable/quantifiable level” Done!. Please put subscript in sulfate Done!. I think it would be easier to analyze the table if samples were grouped by place, instead of being grouped by season. It would be useful to mention in the caption that data were obtained from samples collected in 2016.Answer: Correct. Done. See caption,

Table 2: Is it possible to provide 1-2 sentences in the Methods section to explain how the 2014 data were obtained, and put them as results from the present work instead of “unpublished”? Answer: Data were obtained by chemical analyses of AMD water of S. Domingos that was used for a biorreactor supply in other experiments that I helped to do. I used it for comparison. I don’t agree with the reviewer and I think unpublished data is the correct form to put it.

Please delete the “b.d.l.- below detection limit”, since it was not used in the table. It probably wold be more useful in Table 1. Answer: erased.

Lines 217-218: Sites A1 and A2 does not seem to be affected by AMD. Answer: Correct. Added. Lines 221-22.

Line 223: According to figure 2, the differences between A1 and A2 on one side, and the other samples at the other were even higher, but not mentioned in the text. Answer: Sorry, I do not understand this sentence…

Figures 2-3: Where they build with data obtained from the 2016 samples? Answer: Yes, just table 2 has other sampling dates. Added 2016 in line 225 to clarify.

Lines 258-259: Please re-write the explanation of what cluster is discussed. Answer: Please see new lines 255-256 and line 263 with explanation of which cluster or sub-cluster and its location on the figure.

Lines 261-263: in figure 3, Fe is grouped together with sulfate and conductivity. Please include Fe in the interpretation of clustering. Answer: added in line 268-270.

Line 264: Please explain the meaning of VSC the first time it appears. Answer Please see line 81.

Line 275-276: Table 2 shows obviously different values along time for some parameters, for instance Al, Cu and Fe (L1); conductivity, sulfate, Al, As, Cd, Cu, Mn and Zn for A3; and Fe for S1. Why these differences were not discussed? Can they be explained by changes in the operation of the mining sites? Answer: A3 has the highest concentration of AMD typical parameters, however they cannot be explained due to diferent operation of the mining sites. The explanation is due to being more affected by the degree of waters contamination.

Lines 282-284: Please re-write this sentence. Answer: Done. See new lines 290-293.

Lines 286-289: this paragraph repeats the comment on lines 275-276. The comment for lines 275-276 can be applied to this paragraph as well. Answer: Paragraph 286-289 (new lines 294-298) was re-written. The comments of lines 275-276 just applies for L1 talking about Ca, Mg and Mn.

Line 288: biological data was provided only for 2016. Thus, there is no data in the present work to conclude anything about biological parameters along time. Answer: It is correct, however we present biological information from other studies. The last paragraph was re-written and a new one was added at the end. Please see new Lines 294-309.

Reviewer 2 Report

Acid Mine Drainage effects in the hydrobiology of freshwater streams from three mining areas (SW Portugal): a statistical approach

 Dear sir/ma'am

Thank you for submitting your manuscript " Acid Mine Drainage effects in the hydrobiology of freshwater streams from three mining areas (SW Portugal): a statistical approach" to International Journal of Environmental Research and Public Health, MDPI. I encourage you to write next article.

 No particular mistakes or errors, But:

 Comment:

 A:

1-      Please re-write the abstract ‘An abstract typically contains the following elements’: Some of these points are covered by your current abstract, but some are not.

v  An introduction to the topic of investigation.

v  A motivation for the investigation (or review in this case).

v  A short description of the methodology/tools used.

v  A summary of the results, or in this case the state of the art in the field, advantages, disadvantages, properties and characteristics of the material that have already been investigated, which techniques have been used and what was found.

v  Conclusions of the investigation, or, very importantly in this case, what was not yet investigated, which properties, modifications and applications have not yet been investigated and why it is necessary to consider them.

v  Lastly, what you proposed to do further to develop the field/add new knowledge to the area and what advantages that will bring for yourself and/or other users of the material.

 B: 

1-      Please bold your novelty. What is exactly your novelty?

2-      In an independent Geology map, please specify the Geology layers and between South of Portugal and belong to the Iberian Pyrite Belt (IPB) “Copper belt”.

3-      Please provide details of the classification oxidation-reduction potential (ORP).

4-      Please more explain about Acid Mine Drainage (AMD) methodology and compare different types of statistical distribution in the theory.

5-      Please more Compare the efficiency of your research with some other international research’s, then discuss about it in an independent table”.

6-      Please Use the most recent references from 2020 to 2022, and please correct the quotation reference format.

We are looking forward to hearing from you as soon as possible.

Best Regards,

Reviewing Team

Author Response

Reviewer 3

Acid Mine Drainage effects in the hydrobiology of freshwater streams from three mining areas (SW Portugal): a statistical approach

 Dear sir/ma'am

Thank you for submitting your manuscript " Acid Mine Drainage effects in the hydrobiology of freshwater streams from three mining areas (SW Portugal): a statistical approach" to International Journal of Environmental Research and Public Health, MDPI. I encourage you to write next article.

 No particular mistakes or errors, But:

 Comment:

 A:

1-      Please re-write the abstract ‘An abstract typically contains the following elements’: Some of these points are covered by your current abstract, but some are not.

v  An introduction to the topic of investigation.

v  A motivation for the investigation (or review in this case).

v  A short description of the methodology/tools used.

v  A summary of the results, or in this case the state of the art in the field, advantages, disadvantages, properties and characteristics of the material that have already been investigated, which techniques have been used and what was found.

v  Conclusions of the investigation, or, very importantly in this case, what was not yet investigated, which properties, modifications and applications have not yet been investigated and why it is necessary to consider them.

v  Lastly, what you proposed to do further to develop the field/add new knowledge to the area and what advantages that will bring for yourself and/or other users of the material.

Answer: We really really appreciate the suggestions from the reviewer in how to write an abstract, however due to the 200 words limitation size , it was not possible to add most of the suggestions but we tried to re-write it focusing on the reviewer’s suggestions. Please see new lines 27-31.

 B: 

1- Please bold your novelty. What is exactly your novelty? Answer: organisms affected by AMD, change their communities. Among the organisms that best respond to changes to chemical stress in these extreme environments, are the diatoms, which are the bioindicators per excellence recommended by the Water Framework Directive (WFD 2000/60/C.E.) to evaluate the water courses contamination The most acidophilic species are better prepared to withstand these AMD harsh conditions, and they will develop better, forming communities of tolerant species, being species typical of neutral and alkaline waters absent. Currently, the link between science and economics is very important once an adequate and easy evaluation of mining and industrial impacts facilitates the delimitation of the pollution impacts and can help in the decision of the most appropriate intervention, with reduction of economic and environmental costs, contributing immediately to the quality improvement to sustainable development of agriculture and mining. Thus, we believe that diatoms characterization in this work we will help to establish these links. Please read the new last paragraph of the conclusions.

2-      In an independent Geology map, please specify the Geology layers and between South of Portugal and belong to the Iberian Pyrite Belt (IPB) “Copper belt”. Answer: If the reviewer have the opportunity to read our papers: Luís et al 2009, 2011, 2016, 2019, etc, it will be possible to see all kinds of geological maps with IPB and South portuguese zone included. It was not our objective here to go much further on the geology.

3-      Please provide details of the classification oxidation-reduction potential (ORP).Answer: ORP is the ability of one chemical to oxidize or reduce another chemical. We are in an oxidizing systems so, ORP is positive and as higher is the oxidizing capacity of the waters, higher is the ORP. Please see new lines 270-271.

4-      Please more explain about Acid Mine Drainage (AMD) methodology and compare different types of statistical distribution in the theory. Answer: AMD methodology is explained in water sampling and laboratorial analysis. Please see lines 127-137 and 145-151. Additionaly, we can extend our results and conclusions to all the waters affected by AMD since the Iberian Pyrite Belt contains almost all the sulphide paragenesis of the Iberian Peninsula and is one of the largest sulphide paragenesis in the world.

5-      Please more Compare the efficiency of your research with some other international research’s, then discuss about it in an independent table”. Answer: As the reviewer understands, from our knowledge, our group (Huelva and Aveiro) is the only one that deals with biogeochemistry of AMD affected waters from the IPB.  In this paper, we focused on the main IPB mines from the Portuguese part and compared with our  results of previous papers. And reinforcing that a table (table 2) with previous results of our international team is already included in this paper.

6-      Please Use the most recent references from 2020 to 2022, and please correct the quotation reference format. Answer: Done: please see new paragraph on lines 299-309.

Round 2

Reviewer 2 Report

Acid Mine Drainage effects in the hydrobiology of freshwater streams from three mining areas (SW Portugal): a statistical approach.

Dear sir/ma'am

Thank you for submitting your manuscript " Acid Mine Drainage effects in the hydrobiology of freshwater streams from three mining areas (SW Portugal): a statistical approach" to International Journal of Environmental Research and Public Health, MDPI. Significant changes have occurred in this article (New version). I encourage you to write next article soon.

Yours sincerely,
Reviewing team